# Thyroid Hormones and Spermatozoa: In Vitro Effects on Sperm Mitochondria, Viability and DNA Integrity

**DOI:** 10.3390/jcm8050756

**Published:** 2019-05-27

**Authors:** Rosita A. Condorelli, Sandro La Vignera, Laura M. Mongioì, Angela Alamo, Filippo Giacone, Rossella Cannarella, Aldo E. Calogero

**Affiliations:** Department of Clinical and Experimental Medicine, University of Catania, Via S. Sofia 78, 95123 Catania, Italy; rosita.condorelli@unict.it (R.A.C.); lauramongioi@hotmail.it (L.M.M.); angela.alamo1986@gmail.com (A.A.); filippogiacone@yahoo.it (F.G.); roxcannarella@gmail.com (R.C.); acaloger@unict.it (A.E.C.)

**Keywords:** thyroid hormones, spermatozoa, sperm parameters, sperm mitochondrial potential, sperm motility

## Abstract

The aim of this study wasto assess the in vitro effects of levothyroxine (LT4) on conventional and bio-functional sperm parameters and its implications on fertility. Patients with male idiopathic infertility were enrolled and subjected to examination of the seminal fluid and capacitation according to the WHO 2010 criteria and flow cytometric sperm analysis for the evaluation of bio-functional sperm parameters. LT4 significantly increased the percentage of spermatozoa with high mitochondrial membrane potential (MMP), decreased the percentage of spermatozoa with low MMP and increased sperm motility already at a concentration of 0.9 pmol L^−1^. Therefore, LT4 significantly reduced sperm necrosis and lipid peroxidation ameliorating chromatin compactness. These effects of LT4 were evident at a concentration of 2.9 pmol L^−1^, close to the physiological free-thyroxine (FT4) concentrations in the seminal fluid of euthyroid subjects. We showed a beneficial role of thyroid hormones on sperm mitochondrial function, oxidative stress and DNA integrity. The results of this in vitro study could have a clinical application in patients with idiopathic infertility, clarifying the role of thyroid function on male fertility.

## 1. Introduction

It is known that the thyroid hormones (THs), 3,5,3′-triiodothyronine (T_3_) and thyroxine (T_4_) impact the reproductive function, and thyroid dysfunction is associated with an adverse effect on fertility, both in men [1,2] and women [3]. Indeed, hyperthyroidism or thyrotoxicosis damages spermatogenesis causing maturation arrest, impairs mitochondrial activity and alters the antioxidant systems in rats [4]. Hyperthyroidism results in asthenozoospermia in humans [5,6,7]. In rats, hypothyroidism shows sperm abnormalities partly similar to those reported in hyperthyroidism with an arrest of spermatogenesis, reduction of sperm vitality and an increase of oxidative stress with lipid peroxidation [4,8,9], as well as asthenozoospermia in humans [7]. These semen alterations are reversible both in hypo- and hyperthyroidism and disappear upon achieving euthyroidism [7].

However, the role of THs on male reproductive function and on sperm parameters is still unclear and the scientific evidence is conflicting. To date, all in vitro studies have been performed on animal models; no study has addressed bio-functional and conventional sperm parameters regarding humans.

The presence of thyroid hormone receptors (TR) in rat [10] and human [11] testes suggests a possible role for TH. Moreover, TR are expressed in the whole testis as well as in Sertoli, Leydig and germ cells, epididymis and penis [12].

Thyroid hormones can act with a direct effect on sperm cells as well as on Sertoli and Leydig cells. Moreover, THs act by genomic and nongenomic effects [13].

Genomic effects derive from the binding of T3 with its TR in the nucleus of Sertoli and Leydig cells: it activates gene transcription and protein synthesis, regulating their proliferation and differentiation. T_3_ regulates steroidogenesis through TRα1 in Leydig cells, but Sertoli cells are the main target of T_3_ action in the testes [14].

Nongenomic effects result from the binding of THs with nonnuclear TR in cytoplasmatic membrane, cytoplasm, cytoskeleton and mitochondria where sperm motility is regulated by cAMP pathways.

Some evidences suggest a negative impact of T_4_ on male fertility by showing that it impairs human sperm motility acting with a TR-dependent mechanism [15], decreases the number of spermatozoa/spermatids in the seminiferous tubular lumen and alters the seminal vesicles in animal models [16]. Moreover, T_3_ and T_4_ could damage DNA by increasing reactive oxygen species production [17]. Conversely, other evidences show that free T_4_ concentration seems inversely correlated with sperm DNA damage with a potential protective effect [18] and the treatment with levothyroxine (LT4) increases the weights of both epididymis and testis [19].

The aim of the study was to evaluate whether THs were able to improve the sperm quality (apoptosis, chromatin/DNA integrity and mitochondrial function), to ameliorate oxidative stress, sperm motility and recovery after capacitation.

To accomplish this, the effect of levothyroxine (LT4) on conventional and bio-functional sperm parameters and their implications on fertility were evaluated in vitro.

## 2. Materials and Methods

### 2.1. Patient Selection

The study was conducted in vitro on 15 euthyroid men (mean age 31.2 ± 6.4 years) with idiopathic infertility.

### 2.2. Sperm Analysis and Preparation

Sperm analysis was conducted according to the World Health Organization criteria 2010 [20]. Subsequently, spermatozoa were aliquoted and incubated at 37 °C under 5% CO_2_ with increasing concentrations of LT4 (Sigma-Aldrich S.r.l. Milan, Italy) (0, 0.9, 2.9, 9.9 pmol L^−1^) for 30 min to evaluate the effects on:Sperm progressive motility.Sperm recovery after capacitation by swim-up technique using the Biggers, Whitten, and Whittingham medium (BWW) with capacitating properties. Spermatozoa were then recovered from the supernatant according to their capacity to migrate from the bottom of the test tube to the surface.Bio-functional sperm parameters by flow cytometry: Mitochondrial membrane potential (MMP), vitality, early apoptosis, late apoptosis, necrosis, chromatin compactness, DNA fragmentation and lipid peroxidation (LP).

To date, no study investigated the free-thyroxine FT4 concentration in the semen of men. The LT4 doses used in this study were selected on the basis of the concentrations of FT4 previously tested in the semen of unselected euthyroid men similar to those found in enrolled patients (mean: 3.15 ± 0.7 pmol L^−1^) by chemoluminescence. Moreover, we chose a time of 30 min to reduce the time-dependent sperm damage (reduction of motility, apoptosis, etc.) also considering the analysis times of the seminal fluid, swim-up, etc.

### 2.3. Flow Cytometric Analysis

Flow cytometric analysis was performed using flow cytometer CytoFLEX (Beckman Coulter Life Science, Milan, Italy). The CytoFLEX was equipped with two solid state lasers and six total fluorescence channels (four 488 nm and two 638 nm). We used the FL1 detectors for the green (525 nm), FL2 for the orange (575 nm) and FL3 for the red (620 nm) fluorescence; 100,000 events (low velocity) were measured for each sample. The debris was gated out, by drawing a region on forward versus side scatter dot plot enclosing the population of cells of interest. Computed compensation was made before performing all the analyses. Data were analyzed by the software CytExpert 1.2.

### 2.4. Evaluation of Sperm Apoptosis/Vitality

The externalization of phosphatidylserine (PS) on the outer cell surface is used as an indicator of early apoptosis. The assessment of PS externalization was performed using annexin V, a protein that binds selectively the PS in presence of calcium ions. Therefore, marking simultaneously the cells with annexin V and propidium iodide (PI), we distinguished: Alive (with intact cytoplasmic membrane), apoptotic or necrotic spermatozoa. Staining with annexin V and PI was obtained using a commercially available kit (Annexin V-FITC Apoptosis, Beckman Coulter, IL, Milan, Italy). An aliquot containing 0.5 × 10^6^ mL^−1^ was suspended in 0.5 mL of buffer containing 10 µL of annexin V-FITC and 20 µL of PI and incubated for 10 min in the dark. After incubation, the sample was analyzed by the fluorescence channels 525/40 BP (FITC) and 585/42 BP, 610/20 BP, 690/50 BP (PI). The different pattern of staining allowed us to identify the different cell populations: FITC negative and PI negative indicate alive sperm cells, FITC positive and PI negative indicate spermatozoa in early apoptosis and FITC positive and PI positive indicate sperm cells in late apoptosis.

### 2.5. Evaluation of the Mitochondrial Membrane Potential

The evaluation of MMP was performed using the lipophilic probe 5,5’,6,6’-tetrachloro-1,1’,3,3’tetraethyl-benzimidazolylcarbocyanine iodide (JC-1) which is able to selectively penetrate into mitochondria when it is in monomeric form, emitting at 527 nm. Therefore, JC-1 excitated at 490 nm is able to form aggregates emitting at 590 nm in relation to the membrane potential. When the mitochondrial membrane becomes more polarized, the fluorescence changes reversibly from green to orange. In cells with normal membrane potential, JC-1 is in the mitochondrial membrane in form of aggregates emitting in an orange fluorescence, while in the cells with low membrane potential it remains in the cytoplasm in a monomeric form, emitting a green fluorescence. As regards the sample preparation, we incubated an aliquot containing 1 × 10^6^ mL^−1^ spermatozoa with JC-1 (JC-1 Dye, Mitochondrial Membrane Potential Probe, Labochem Science, Catania, Italy) for 10 min, at a temperature of 37 °C and in the dark; after incubation, the cells were washed in PBS and analyzed (FL1 and FL3) obtaining two different populations:spermatozoa with high mitochondrial membrane potential (H-MMP)spermatozoa with low mitochondrial membrane potential (L-MMP).

### 2.6. Assessment of DNA Fragmentation

The evaluation of DNA fragmentation was performed by the TUNEL assay. This method uses terminal deoxynucleotidyl transferase (TdT), an enzyme that polymerizes at the level of DNA breaks, modifying nucleotides conjugated to a fluorochrome. The TUNEL assay was performed using a commercially available kit (Apoptosis Mebstain kit, DBA s.r.l, Milan, Italy). To obtain a negative control, TdT was omitted from the reaction mixture; the positive control was obtained by pre-treating spermatozoa (about 0.5 × 10^6^) with 1 mg/ml of deoxyribonuclease I, not containing RNAse, at 37 °C for 60 min prior to staining. After fixation (4% paraformaldehyde) and permeabilization of the plasma membrane (ethanol 70%), we incubated an aliquot containing 0.5–1 × 10^6^ mL^−1^ spermatozoa for 60 min, at a temperature of 37 °C; after incubation, the cells were washed in PBS and analyzed (FL1).

### 2.7. Degree of Chromatin Compactness Assessment

Chromatin compactness assessment was evaluated after a process of cell membrane permeabilization; in this way fluorophore was able to penetrate the nucleus. An aliquot of 1 × 10^6^ spermatozoa was incubated with LPR DNA-Prep Reagent containing 0.1% potassium cyanate, 0.1% NaN3, non-ionic detergents, saline and stabilizers (Beckman Coulter, IL, Milan, Italy) in the dark at room temperature for 10 min. It was then incubated with Stain DNA-Prep Reagent containing 50 µgm L^−1^ of PI (<0.5%), RNase A (4 KUnitz/mL), <0.1% NaN_3_, saline and stabilizers (Beckman Coulter, IL) in the dark at room temperature for 30 min. The cells were analyzed (FL3).

### 2.8. Sperm Membrane Lipid Peroxidation Evaluation

LP evaluation was performed using the probe BODIPY (581/591) C11 (Invitrogen, Thermo Fisher Scientific, Eugene, Oregon, USA), which responds to the attack of free oxygen radicals changing its emission spectrum from red to green after being incorporated into cell membranes. This change of the emission is detected by the flow cytometer which provides an estimate of the degree of peroxidation. About 2 × 10^6^ of spermatozoa were incubated with 5 mM of the probe for 30 min in a final volume of 1 mL. After washing with PBS, flow cytometric analysis was conducted using the 525/40 BP (FITC) and 585/42 BP (PE) fluorescence channels.

This study was approved by the Ethics Committee of University teaching Hospital of Policlinico-Vittorio Emanuele, University of Catania (Catania, Italy).

All methods were performed in accordance with the relevant guidelines and regulations. All participants were asked for and provided their informed consent.

### 2.9. Statistical Analysis

The results are expressed as mean±SEM throughout the study. Statistical analysis of the data was performed using Student’s t-test, when possible, and by one-way analysis of variance (ANOVA) followed by the Duncan’s Multiple Range test. SPSS 22.0 for Windows was used for statistical analysis (SPSS Inc., Chicago, USA). Results with a *p*-value less than 0.05 were accepted as statistically significant.

## 3. Results

The conventional sperm parameters (mean ± SEM) of enrolled men are shown in Table 1. We reported also the seminal FT4 measurement of the same patients in Table 1.

Effects on sperm motility: LT4 significantly increased (10%) sperm progressive motility at a concentration of 0.9 pmol L^−1^ compared to LT4 0, whereas LT4 decreased (9% and 10% respectively) sperm motility at concentrations of 2.9 and 9.9 pmol L^−1^ (*p* < 0.05 vs. LT4 0) (Figure 1).

Effects on bio-functional sperm parameters:LT4 significantly increased the percentage of spermatozoa with high MMP and decreased the percentage of spermatozoa with low MMP. This effect was similar for all the three concentration of LT4 tested (*p* < 0.05 vs. LT4 0) (Figure 2). Moreover, LT4 reduced the percentage of spermatozoa in necrosis at the concentration of 2.9 and 9.9 pmol L^−1^ (*p* < 0.05 vs. LT4 0) while chromatin compactness already improved to a concentration of 0.9 and became statistically significant at the concentration of 9.9 pmol L^−1^ (Figure 3). Finally, LP improved already at a concentration of 0.9 pmol L^−1^ (*p* < 0.05 vs. LT4 0) (Figure 3). DNA fragmentation did not vary for all the three concentration of LT4 tested. Non-significant data are shown in Table 2.

Effects on sperm recovery: The number of spermatozoa recovered by capacitation did not change significantly after incubation with LT4 though it increased at a concentration of 0.9 pmol L^−1^ compared to LT4 0 (Table 2).

The anthropometric parameters and hormone measurements in serum (mean ± SEM) of enrolled patients are shown in Table 1.

## 4. Discussion

Many aspects of the possible role of THs are yet to be clarified. Our study investigated the in vitro effects of LT4 on human sperm motility and bio-functional sperm parameters at physiological concentrations as measured in the seminal fluid, for the first time. Accordingly, we selected a physiological dose (2.9 pmol L^−1^) of LT4 and concentrations respectively lower (0.9 pmol L^−1^) and higher (9.9 pmol L^−1^) for our in vitro experiments.

We hypothesized a relationship between sperm motility and mitochondrial function as previously described [21,22,23]. Our results showed that MMP improved significantly after incubation with LT4 already at a concentration of 0.9 pmol L^−1^. Similarly, LT4 was able to increase sperm motility at the same concentration, but at higher concentrations than 0.9 pmol L^−1^, sperm progressive motility decreased significantly.These data confirmed the relationship between these two spermatic parameters, so we hypothesized that LT4 improving MMP also stimulated sperm motility until substrates are exhausted with a subsequent decline in motility itself.

Xian and colleagues showed a negative impact of LT4 on human sperm motility at a concentration of 10 μM [15]. These findings confirm our results, allowing us to understand that high concentrations, which simulate an in vitro condition of hyperthyroidism, damage spermatozoa.

Moreover, the decreased sperm motility observed with the two highest concentrations of LT4 used in the present study may be due to an excessive consumption of substrates after exposure to higher concentrations of LT4 (greater than 0.9 pmol L^−1^). Indeed, LT4 improved MMP by increasing oxidative phosphorylation and sperm motility related to this biochemical mechanism until no further substrates were available. A condition of hyperthyroidism alters cytochrome c oxidase activity decreasing mitochondrial activity [4]. In addition, the number of spermatozoa recovered by swim-up increased only at the concentration of 0.9 pmol L^−1^, and did not improve at higher concentrations due to the concomitant reduction in sperm motility. Based on our findings, we hypothesize that slight additions of LT4, in a seminal fluid with constant concentrations of LT4 as in euthyroid men, may result in improved sperm motility. If the concentration of LT4 exceeds this level, these parameters worsen as in a clinical situation of hyperthyroidism due to excessive consumption of the substrates.

Therefore, we showed a beneficial effect of LT4 on sperm oxidative stress that ameliorates after incubation with 0.9 pmol L^−1^. This effect may explain the reduction in the percentage of spermatozoa in necrosis or with altered chromatin compactness after incubation with LT4.

The in vitro effects of LT4 on viability, chromatin compactness and LP can be explained by the presence of the nuclear TR in spermatozoa [14]. Dobrzynska et al., suggested that T_3_ and T_4_ could damage DNA by increasing reactive oxygen species production at the concentration of 80 μM [17] but these concentrations are more elevated compared to the physiological doses found in the seminal fluid. Conversely, a condition of hypothyroidism impairs the antioxidant defense system and testicular physiology [9]. We showed that, near physiological concentrations, LT4 improved oxidative stress without altering sperm DNA fragmentation.

Therefore, slight additions of seminal LT4 could be useful for improving the semen sample so that it could be used for assisted reproductive techniques.

## 5. Conclusion

We showed a role of THs on sperm mitochondrial function. In addition, LT4 seems to act by decreasing the oxidative stress and improving DNA integrity. Therefore, we could achieve a beneficial effect on both conventional and bio-functional sperm parameters maintaining thyroid hormone concentrations sufficient and mostly constant in the seminal fluid. The clinical application of these data could be useful in some cases of idiopathic infertility. Further studies should be undertaken to better investigate the intracellular and biomolecular mechanisms supporting these findings. The next step on this topic could be to investigate hypothyroid infertile and euthyroid fertile men to better understand the role of thyroid hormones on male reproductive function.

## Figures and Tables

**Figure 1 jcm-08-00756-f001:**
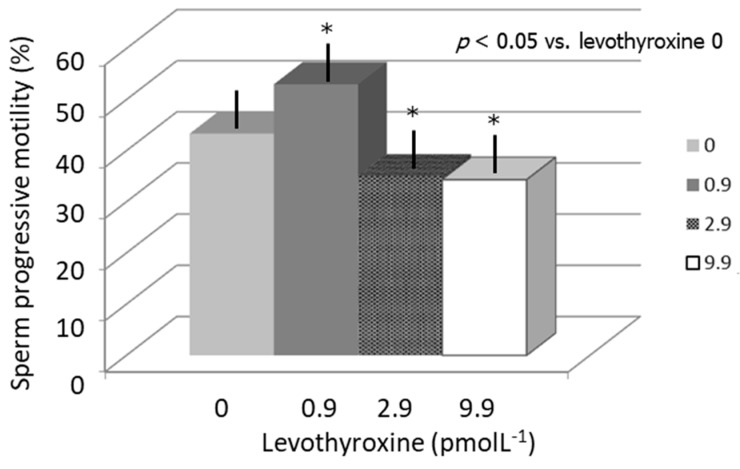
Sperm progressive motility after incubation with increasing concentrations of levothyroxine (LT4).

**Figure 2 jcm-08-00756-f002:**
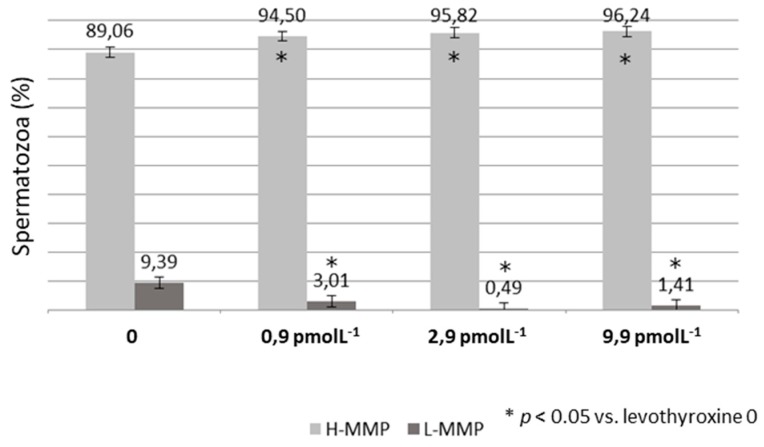
Percentage of spermatozoa with high (H-MMP) and low (L-MMP) membrane mitochondrial potential after incubation with increasing concentrations of levothyroxine (LT4).

**Figure 3 jcm-08-00756-f003:**
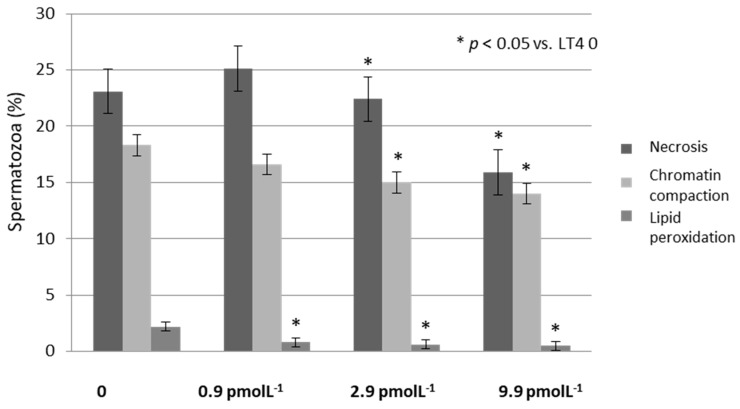
Sperm necrosis, chromatin compactness and lipid peroxidation after incubation with increasing concentrations of levothyroxine (LT4).

**Table 1 jcm-08-00756-t001:** Conventional sperm parameters, FT4 seminal measurements, anthropometric parameters and hormone measurements in serum (mean ± SEM) of enrolled men.

	Patients (n = 15)
Concentration (10^6^ mL^−1^)	42.80 ± 12.0
Total count (10^6^ ejaculate^−1^)	107.00 ± 30
Progressive motility (%)	39.80 ± 3.3
Total motility (%)	77.90 ± 9.8
Normal form (%)	8.30 ± 0.8
Leukocytes (10^6^ mL^−1^)	0.70 ± 0.2
Seminal FT4 (pmol L^−1^)	3.15 ± 0.7
Age	31.2 ± 6.4
BMI (kg m^−2^)	24.2 ± 1.3
Waist circumference (cm)	94.0 ± 2.4
TSH (μUI m L^−1^)	2.1 ± 0.5
FT4 (pmol L^−1^)	10.2 ± 0.9
FT3 (pmol L^−1^)	4.7 ± 0.8
FSH (UI L^−1^)	2.3 ± 0.1
LH (UI L^−1^)	2.4 ± 0.2
Total testosterone (ng mL^−1^)	6.1 ± 1.1

**Table 2 jcm-08-00756-t002:** Bio-functional sperm parameters and sperm recovery after incubation with increasing concentrations of levothyroxine (LT4).

Sperm Parameters	LT4 Concentrations			
	0	0.9 pmol L^−1^	2.9 pmol L^−1^	9.9 pmol L^−1^
Alive	65.2 ± 4.6	53.6 ± 8.8	57.3 ± 6.2	54.0 ± 9.3
Early Apoptosis	2.0 ± 0.8	1.5 ± 0.5	1.6 ± 0.7	1.8 ± 0.5
Late apoptosis	18.6 ± 8.6	19.9 ± 8.8	15.2 ± 6.4	23.9 ± 7.8
DNA fragmentation	3.4 ± 0.5	3.8 ± 0.6	4.1 ± 0.7	3.9 ± 0.7
Sperm recovery by swim-up	18.7 ± 8.8	21.3 ± 10.0	14.9 ± 7.7	11.9 ± 10.8

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
