# Peer review of "Thyroid Hormones and Spermatozoa: In Vitro Effects on Sperm Mitochondria, Viability and DNA Integrity"

_jcm, 2019, doi:10.3390/jcm8050756_

Reviewer 1 Report

Thank you for an interesting manuscript addressing topics of interest. Any knowledge on in vitro treatment for improvement of quality og ejaculated spermatozoa is welcome.

Overall, my impression is that the English language could be improved in several points, especially regarding singular/plural forms.

Included are some additional comments and suggestions for improvement and clarifying of the scientific content.

Line 59: It could have been more clearly explained that the effect was tested directly on spermatozoa (in vitro), not on patients/donors, even if these were stated as euthyroid. Consider to mention this already in the title.

Line 74: At what temperature and atmospheric conditions were the recovered spermatozoa incubated?

Line 80: Unclear.

"unselected euthyorid men" should be replaced by another expression if Table 1 (as stated in line 153) reflects the FT4 concentration in the semen samples from the infertile 15 men in the study, prior to sample incubation.

What is the reference value of FT4 in human seminal plasma? (Reference paper or review should be included).

Additional value would be added to the paper if the background for the choice of concentrations of LT4 in the trial was explained, maybe based on literature? (Ref 19 refers to oral intake of T4 in rats…)

 Also, the choice of incubation time with LT4 might be explained or discussed since both a incubation time x dosage interaction might be expected, plus the impact of 30 minutes’ incubation itself upon (progressive) motility and possibly other parameters might not be negligeable.

Line 117: Treatment of positive and negative control clearly described, but how were the samples treated?

Line 141: New paragraph/subtitle, or re-allocate these lines. They do not belong under "Sperm membrane lipid peroxidation".

Line 146: Phrase in duplicate.

Line 152: Referring to Table 1, a description of method of assessment of progressive motility (and total motility) is lacking in Materials and Methods.

Line 168:«hormone serum measurements» could be changed to «hormone measurements in serum» (also valid for title of Table 3, line 288).

Line 204: PMA techniques: This acronym should be explained or written in full length.

Line 273: The figure does not reflect the membrane potentials, but the percentage of spermatozoa with H-MMP and L-MMP (which are not explained in the legend).

Line 282/Table 1: «mil/ml» and «mil/ejaculate» are not valid units, the author is encouraged to use 106 instead of «mil».

Line 285/Table 2: Sperm recovery – if sperm were aliquoted and subsequently treated after swim-up, the background for this table row needs to be explained. Was there a second swim-up treatment after the incubation with LT4?

Line 288/Table 3: Title «hormone serum measurements» could be changed to «hormone measurements in serum».

Author Response

Attached the rebuttal Letter

Reviewer 2 Report

The study by Condorelli et al. showed a role of thyroid hormones on human sperm mitochondrial function.

The topic of the study is very interesting, since it provides the first evidence of a beneficial role of thyroid hormones on sperm mitochondrial function, oxidative stress and DNA integrity.  

Minor points:

Methods:

- The paragraph “Evaluation of the mitochondrial membrane potential” should be modified according to Figure 2 (high and low MMP).

Results:

- Please indicate the percentage of increase/decrease of measured parameters after incubation with increasing concentrations of levothyroxine.

- Table 1 and Table 3 could be combined in a single table (new Table 1).

Discussion:

- The sentence: “We hypothesized a relationship between sperm motility and mitochondrial function as previously described [21]” is not completely supported by the results shown in Figures 1 and 2. In fact, the observed decrease in sperm motility (0.9 and 9.9 pmol/L LT4) is not associated to a parallel decrease in membrane mitochondrial potential, which shows similar values at all tested concentrations of LT4. 

Authors should modify lines 176-180 in order to better discuss their results starting from the known relationship between sperm motility and mitochondrial functionality (Paoli et al., 2011; Ferramosca et al. 2012; Moscatelli et al. 2017).

Author Response

Attached the rebuttal Letter

Round  2

Reviewer 1 Report

Thank you for performing the requested changes and clarifications. The only comment I have now is about the % decrease/increase in sperm motility; lines 164-165. Should it be % or % units?

Reviewer 2 Report

As far as I am concerned, the manuscript is now acceptable to be published.